# Adaptive Gene Expression Induced by a Combination of IL-1β and LPS in Primary Cultures of Mouse Astrocytes

**DOI:** 10.3390/cells14211737

**Published:** 2025-11-05

**Authors:** Thierry Coppola, Gwénola Poupon, Hélène Rangone, Stéphane Martin, Patricia Lebrun

**Affiliations:** Université Côte d’Azur, CNRS, Inserm, IPMC, Sophia Antipolis, F-06560 Valbonne, France; gpoupon@ipmc.cnrs.fr (G.P.); helenerangone@gmail.com (H.R.); martin@ipmc.cnrs.fr (S.M.)

**Keywords:** astrocyte, inflammation, glutamate, lactate, solute carriers

## Abstract

Astrocytes are vital cells within the central nervous system (CNS), as they perform a critical role in supporting neurons by providing nutrients, such as lactate for energy, and safeguarding them against the toxicity of excessive neurotransmitters, such as glutamate. This study investigates astrocyte adaptive mechanisms in response to chronic inflammation. The primary aim is to assess the long-term effects of an inflammation-induced environment using a combination of lipopolysaccharide (LPS) and interleukin-1β (IL-1β), on the expression of key genes involved in essential metabolic pathways for astrocyte function, including glutamate metabolism and clearance, lactate synthesis and transport, and glucose metabolism. We observed an upregulation of the glutamate transporter *eaat2* (but not *eaat1*), leading to glutamate accumulation and altered glutamate-glutamine cycling, as well as increased glycolytic activity and lactate production/export via hexokinases (*hk1* and *hk2*) and the *mct4* lactate transporter. Interestingly, these mechanisms are reversible, indicating a precisely controlled adaptive system. This investigation facilitated the identification of the signaling pathways involved in astrocyte adaptive responses to stress. This will further guide our investigations towards the more complex domain of resistance and adaptation of CNS in pathophysiological conditions.

## 1. Introduction

The central nervous system (CNS) is a complex organ composed of various cell types. Among them, neurons are specialized for brain communication and signal transmission, and astrocytes closely support their function. These glial cells help to maintain neuronal health by delivering essential nutrients and metabolic support. Astrocytes thus act as a metabolic hub, supplying nearby neurons with compounds, mainly lactate, produced by glycolysis and serving as the primary energy substrate for neurons. Unlike glucose, which plays a limited role due to its moderate interstitial concentration [1,2], lactate can be directly metabolized by neurons through mitochondrial respiration [3]. In brief, astrocytes uptake glucose via GLUT1 transporters and metabolize it to produce lactate via LDH. Then, they export lactate through MCT1/4 for neuronal uptake (MCT2) in the astrocyte–neuron lactate shuttle, providing energy supply to neuronal cells.

Among the various molecules provided by astrocytes, amino acids serve as precursors to neurotransmitters. For example, glutamine is the precursor of the primary excitatory neurotransmitter, glutamate, as well as GABA, the main inhibitory neurotransmitter [4,5]. The synthesis of many neurotransmitters occurs through specific metabolic pathways that supply these simple precursors. Astrocytes take up synaptic glutamate via EAAT1/2 transporters and convert it into glutamine using glutamine synthetase for neuronal recycling. This synaptic glutamate uptake by astrocytes not only provides resources for glutamine regeneration but also removes excess neurotransmitters from the synaptic cleft [6], as their accumulation could make the synapse silent [7]. In the case of glutamate, the role of astrocytes extends significantly beyond synapse silencing, as this acidic amino acid also induces rapid neuronal apoptosis. The cytotoxicity of glutamate and its contribution to neuronal dysfunction are now extensively documented. By preventing the accumulation of glutamate around neurons, astrocytes serve a protective function and promote neuronal survival [8]. Enhanced expression of EAAT2 has been reported following ischemic injury, but also during the progression of neurodegenerative diseases [9]. These processes collaborate to ensure neuroprotection and energy homeostasis. Investigating the roles of astrocytes in glutamate metabolism and clearance, lactate synthesis and transport, and glucose metabolism is essential for advancing our comprehension of the intricate relationships between brain function and disease. Impairments in neuroprotection and dysregulation of CNS function contribute to various neuropathological conditions, including Alzheimer’s disease, Parkinson’s disease, and Huntington’s disease, among others. The majority of these neurodegenerative disorders are closely linked with chronic low-grade neuroinflammation, and numerous research initiatives are underway to elucidate the origins and implications of this phenomenon [10]. Whereas neuroinflammation was previously recognized as one of the most detrimental consequences of neurodegenerative disease and protein aggregate formation, it is increasingly evident that this exacerbated inflammation also contributes to the onset of such pathologies.

In this study, primary astrocyte cultures are used to evaluate the long-term effects of exposure to lipopolysaccharide endotoxin (LPS) and IL-1β, two molecules associated with neuroinflammation. IL-1β is a crucial cytokine that triggers a reactive phenotype in astrocytes; however, its effects are notably species-dependent. In human primary astrocytes, IL-1β (typically 10 ng/mL) is a potent activator, precipitating significant and irreversible morphological alterations, such as a stellate shape, within 6 to 24 h [11]. These changes result from the deactivation of the Rho GTPase–ROCK pathway, causing cytoskeletal reorganization and limiting cell migration [12]. IL-1β also induces a pronounced pro-inflammatory response in human astrocytes, upregulating genes such as tumor necrosis factor alpha (TNF-α) and inducible nitric oxide synthase (iNOS) at concentrations as low as 10 pg/mL. While IL-1β can activate the NF-κB transcription factor within 20 min in a rat glioma cell model, prolonged exposure (48–72 h) to IL-1β in human astrocytes results in cellular impairment and apoptosis, involving the activation of cleaved caspase-3 [13]. Conversely, IL-1β alone has only a limited and sporadic effect on primary mouse astrocytes, notably failing to induce TNF-α or iNOS production. Nevertheless, prior research demonstrates that, despite its modest individual impact on mouse astrocytes, IL-1β can enhance the astroglial response to other stimuli [14]. LPS has been extensively used as a biomarker for systemic inflammation, which is notably associated with conditions such as obesity, diabetes, and colorectal cancer. Via its action on the Toll-like Receptor-4 (TLR4), LPS functions as an inducer of neuroinflammation, resulting in elevated levels of pro-inflammatory cytokines. Consequently, LPS treatment elicits a robust inflammatory response in primary astrocyte cultures, characterized by increased GFAP expression and hypertrophy [15]. Concentrations ranging from 0.1 µg/mL to 100 µg/mL effectively stimulate astrocytes, leading to the biosynthesis and secretion of pro-inflammatory mediators such as IL-1β, IL-6, and TNF-α within 8 to 48 h. This inflammatory response reduces cell viability and proliferation while concurrently increasing cell migration and necrotic cell death. Furthermore, LPS enhances the transport function of P-glycoprotein (P-gp) and activates the NF-κB pathway by facilitating P65 nuclear translocation [15]. These effects are often dose-dependent, with higher concentrations (e.g., 10–100 µg/mL) inducing a more significant inflammatory response over time. These investigations concentrate on the adaptive mechanisms of astrocytes in response to chronic inflammation. Given that IL-1β and LPS are capable of activating both convergent and non-redundant signaling pathways [16,17], the primary objective of this study is to assess the long-term effects of inflammation induced by a combination of IL-1β and LPS on the expression of key genes involved in essential metabolic pathways for astrocyte function, including glutamate metabolism and clearance, lactate synthesis and transport, and glucose metabolism. The research aims to characterize the autonomous adaptive responses of astrocytes within a pro-inflammatory environment and to validate the use of primary astrocyte cultures as a pertinent model for such investigations.

## 2. Materials and Methods

### 2.1. Mouse Line

C57BL/6 female WT mice were purchased from Janvier (St Berthevin, France). All animals were handled and treated in accordance with the European Council directives for the Care and Use of Laboratory Animals and following the ARRIVE guidelines. Mice had free access to water and food and were exposed to a 12 h light/dark cycle. The temperature in the accommodation area was maintained at 23 ± 1 °C, with a relative humidity of 45–65%. The primary culture was performed using exclusively newborn pups from in-house-bred pregnant mice (10–20 weeks old). Although this study requires no experimentation on living animals, we are following the rules under the EU regulation 2010/63/EU for the housing, manipulation and euthanasia of animals.

### 2.2. Ethics Committee

The animal study protocol was approved by the National Animal Care and Ethics Committee (project reference APAFIS #18648-201901111154666 v6, authorization valid until 10 December 2028).

### 2.3. Preparation of Primary Astrocyte Cultures

Neonatal mouse brains (P0–P3) are extracted, bathed for 10 s in 70% ethanol, and placed in Hank’s Balanced Salt Solution (HBSS) at room temperature (RT). The meninges, striatum, hippocampus, and olfactory bulbs are removed, leaving only the cortices. After two washes in HBSS, the cortices are mechanically dissociated by pipetting. Cells (10^5^/well) are then seeded on poly-lysine (0.2 mg/mL)-coated 24-well plates, or T75 flask (6 × 10^6^ cells/flask) in Dubelcco’s Modified Eagle Medium (DMEM) 4.5 g/L glucose supplemented with 2 mM L-glutamine, 50 UI/mL penicillin, 50 µg/mL streptomycin, 10% fetal calf serum (FCS), 10% horse serum (HS) and placed at 37 °C under 5% CO_2_ for 24 h.

### 2.4. Cell Treatments

Astrocytes were treated with the pro-inflammatory cytokine IL-1β at 10 ng/mL, LPS at 1 µg/mL, and a saturated fatty acid. These treatments were applied for 2 or 5 days as indicated.

### 2.5. Protein Extraction

Cells were lysed with 100 µL of lysis buffer (10 mM Tris-HCl, 150 mM NaCl, 0.5 mM EDTA, pH8.0, 1.5% CHAPS) and frozen at −20 °C. Samples were then centrifuged (14,000 rpm, 10 min at 4 °C), and 80 µL of the supernatants were collected. Protein concentration was determined using BioRad (Marnes-la-Coquette, France) assays (Bradford assay) according to the manufacturer’s instructions.

### 2.6. LDH Activity

Cytosolic LDH activity (Cytotoxicity detection kit, Roche (Boulogne-Billancourt, France, #11644793001) was measured by monitoring absorbance at 492 nm during the conversion of tetrazolium to the colored formazan. After the indicated pharmacological treatments, cells were lysed with lysis buffer, and 20 µL aliquots were loaded into wells of a microtiter plate containing 80 µL sterile water. Next, 100 µL of reaction mix (diaphorase, NAD+, INT, lactate, sodium) was added to each well at RT. After a 20 min incubation time at 37 °C, absorbance was read on a plate reader (Multiskan FC, Thermo Scientific, Courtaboeuf, France) at 492 nm.

### 2.7. Cell Viability Assay

Cell viability was assayed using the “CellTiter Glo Luminescent cell viability assay” from Promega #G7570 (Promega, Charbonnières-les-Bains, France), according to the manufacturer’s protocol.

### 2.8. Quantification of Cytokines IL-6 and TNF-α

Cell lysate (2 µL) is added to 8 µL of reaction buffer (Immunoassay buffer 10×, PerkinElmer, Villebon-sur-Yvette, France) containing acceptor bead-linked anti-cytokine antibodies (1/500) and biotinylated anti-cytokine antibodies (1/500) in an OptiPlate-384 plate (PerkinElmer, Villebon-sur-Yvette, France). After 1 h at RT under gentle agitation, 10 µL mix containing streptavidin-coated donor beads (1/125) was added to the reaction and incubated at RT under gentle agitation for 30 min. In parallel, a standard range is run with IL-6 and TNF-α. The plate is then read using Enspire Alpha (PerkinElmer, Villebon-sur-Yvette, France).

### 2.9. Glutamate/Glutamine Assay

To measure glutamine and intracellular glutamate concentrations, we used the Glutamine/Glutamate-Glo™ Assay (Promega, Charbonnières-les-Bains, France, #J8021). 12.5 µL of cell lysate was mixed with 12.5 µL of glutaminase solution for the glutamate and total glutamine samples. In comparison, 12.5 µL of buffer solution was added to the glutamate samples in 96-well plates with a clear bottom. 25 µL of glutamate detection solution (Luciferin detection solution, reductase, reductase substrate, and NAD+) was added to all wells. Absorbance was measured with a luminometer (GloMax, Promega Charbonnières-les-Bains, France).

### 2.10. Lactate Assay

To assess lactate accumulation in the extracellular medium, we used the Lactate-Glo Assay from Promega, Charbonnières-les-Bains, France, (#J5021). Culture medium was collected at 0 and 5 days after treatment. 25 µL of extracellular medium was mixed with 25 µL reagent detection solution (Promega) containing 0.25 µL reductase, 0.25 µL reductase substrate, 0.25 µL LDH and 0.25 µL NAD. After 1 h at RT, the amount of NADH produced was measured with a Luminometer (GloMax, Promega, Charbonnières-les-Bains, France).

### 2.11. Reverse Transcription and Real-Time PCR Analysis

Total RNAs from astrocytes were isolated using a phenol–chloroform extraction method [14] with the TRI Reagent buffer (Sigma Aldrich, St Quentin Fallavier, France, #T9424), according to the manufacturer’s instructions. Putative contaminating genomic DNA was removed with the Turbo DNase kit (Invitrogen, Courtaboeuf, France). First-strand cDNA was then synthesized from 2 μg total RNA using the Superscript IV reverse transcriptase (Invitrogen, Courtaboeuf, France, #18091050). Real-time PCR assays (duplicates for each condition) were performed in a 96-well plate on a LightCycler 480. The Ct value of each gene of interest (Ctgoi) was normalized to the average of the Ct of the reference (Ctref) genes: ΔCt = Ctgoi − Ctref, with Ctref = (Ctref1 + Ctref2) × (1/2). ΔΔCt = ΔCt experimental condition − ΔCt control condition. Values were expressed as 2^−ΔΔCt^ normalized to untreated cells as a control. The 18S gene was used as a reference. (Primer sequences can be downloaded as Appendix A).

### 2.12. Immunostaining

Cells were washed with phosphate-buffered saline (PBS) and then fixed with 4% paraformaldehyde in PBS at RT for 20 min. Cells were next permeabilized, blocked with 3% HS in PBS containing 0.1% Triton X100 for 20 min, and incubated with primary antibodies for 2 h at RT: anti-GFAP antibodies (1:500, #Z0334, Dako, Les Ulys, France) and anti-EAAT2 antibodies (1/500; #AB1783, Millipore, Sigma Aldrich, St Quentin Fallavier, France). After three washes in PBS, secondary antibodies conjugated to Alexa Fluor 488 (GFAP and EAAT2) or Alexa647-Phalloïdine labeling polymerized actin cytoskeleton were incubated for 1 h at RT before mounting in Mowiol (#9002-89-5, Sigma Aldrich, St Quentin Fallavier, France) and imaging.

### 2.13. Immunolabelling Quantification

Quantification of total immunolabelling from immunostaining was performed using ImageJ software v1.54p. All the recorded images were analyzed using the same set of measurements (SP5 confocal system from Leica, France, Paris). The total area fluorescence measured was divided by the number of cells to obtain a fluorescence in arbitrary units (AU)/cell, as described on the ImageJ website (https://imagej.net/ij/docs/index.html) (accessed on 29 October 2025) and [18]. The mean ± SEM across 6 acquisitions per condition is shown.

### 2.14. Statistical Analysis

Statistical analyses were calculated using GraphPad Prism v7.03 (GraphPad Software, Inc USA, RRID: SCR_002798). Data were expressed as mean ± SEM. One-way ANOVA was performed with a Tukey post hoc test for multiple comparison data sets, and a Ratio paired t-test was used for the qPCR analysis and LDH assays. For each test, the homogeneity of variance (variance between groups is approximately equal) was verified, as the F value from the ANOVA test was consistently well above 1. *p* < 0.05 was considered significant.

## 3. Results

To simulate a cellular stress similar to neuroinflammation, we exposed primary astrocytes to key pro-inflammatory triggers, IL-1β and LPS, at two time points. Figure 1 illustrates the experimental protocol, detailing the timing of treatments and sample collection used to analyze the impact of inflammation on the astrocytic physiological functions. Astrocytes isolated from newborn mice were incubated for either 2 or 5 days with IL-1β, LPS, or a combination of both.

To determine whether these treatments trigger an inflammatory response, we measured both intracellular and extracellular levels of two key cytokines: Interleukin-6 (IL-6) and Tumor Necrosis Factor-alpha (TNF-α). Treatment with LPS alone or combined with IL-1β significantly increased the expression and secretion of IL-6 and TNF-α. After two days of exposure, IL-6 levels rose both in the extracellular medium (Figure 2A) and inside the cells (Figure 2C), and this increase persisted after five days of incubation (Figure 2B,D). Similarly, TNF-α levels increased as early as day 2, both outside the cells (Figure 2E) and inside (Figure 2G), remaining elevated after five days (Figure 2F,H). Interestingly, incubation with IL-1β alone did not significantly alter IL-6 or TNF-α levels but appeared to enhance LPS effects in several conditions (intracellular IL-6 and TNF-α after 5 days of treatment). Based on these findings, we chose the IL-1β/LPS co-treatment for both 2 and 5 days to activate LPS and IL-1β pathways simultaneously, capable of eliciting physiologically relevant astrocytic responses.

As a first step, we assessed cell viability and survival under pro-inflammatory treatments. We initially measured the total protein content as an indicator of cell loss. Except for a slight increase in protein content in the LPS condition, we observed the preservation of the cell population (Figure 3A). To confirm cell viability, we measured cellular respiration by assessing ACT activity under different conditions. Despite some variability, there was no significant loss of viability (Figure 3B,C), indicating that the treatments did not cause substantial cell death. To further examine potential morphological changes in astrocytes under IL-1β/LPS-induced stress, we imaged the subcellular distribution of specific cytoskeletal markers. Treated cells showed membrane remodeling, indicated by the distribution of the major astrocytic intermediate filament GFAP, especially after 2 and 5 days of treatment (Figure 3D). Additionally, the actin cytoskeleton, visualized with fluorescent phalloidin, revealed changes in astrocytic morphology following IL-1β/LPS treatment (Figure 3E). Overall, these findings support that IL-1β/LPS treatments induce a reorganization of astrocytic morphology, while cell viability remains largely unaffected.

Given the critical role of astrocytes in regulating glutamate homeostasis, we focused our gene expression analysis on genes encoding proteins involved in both extracellular and intracellular glutamate transport. After two days of IL-1β/LPS treatment, the level of mRNAs encoding the glutamate transporter EAAT2 was significantly increased (1.055 vs. 2.322 at day 2), and this upregulation persisted after 5 days (1.055 vs. 2.161 at day 5; Figure 4A). In contrast, the mRNA expression of the other glutamate transporter, EAAT1, remained unchanged (Figure 4B), indicating a selective upregulation of EAAT2.

We next assessed the expression of GLUD, the enzyme that converts glutamate into α-ketoglutarate, a key intermediate in the tricarboxylic acid (TCA) cycle. Although there was a slight trend toward an increase, *glud* mRNA levels were not significantly affected by the treatment (Figure 4C). This result suggests that increased intracellular glutamate is not being redirected toward enhanced energy production through the TCA cycle. Finally, we measured the intracellular concentration of glutamate following IL-1β/LPS treatment (Figure 4D). After 5 days of treatment, during which the IL-1β/LPS-induced stress was maintained, there was a threefold increase in the intracellular concentration of Glu (8.12 μM in control vs. 25.80 μM). To confirm the transcriptional data for *eaat2*, we examined the levels of EAAT2 protein using immunostaining. EAAT2 immunoreactivity was significantly increased, particularly after 5 days of IL-1β/LPS treatment (Figure 4E,F).

Altogether, these data reveal that IL-1β/LPS-induced stress specifically upregulates EAAT2, but not EAAT1, in primary cultured astrocytes. This correlates with a substantial accumulation of intracellular glutamate, suggesting selective activation of the glutamate uptake pathway.

Since the cytosolic concentration of glutamate depends on both its uptake and its metabolic conversion, we measured the mRNA expression level of glutamate ammonia ligase (*glul*), which is required for synthesizing glutamine from cytosolic glutamate. After two days of treatment, we observed a significant decrease in *glul* mRNA transcripts, which persisted over the 5 days of exposure to the IL-1β/LPS factors (Figure 5A; 0.99 in the control vs. 0.65 at day 2 and 0.74 at day 5). The glutamine produced by astrocytes plays a key role in the glutamate-glutamine cycle, which ensures the recycling of glutamate released by neurons. Glutamine is exported from astrocytes to neurons via astrocyte-specific transporters, SNAT3 being the major transporter. Interestingly, our analysis reveals that the *snat3* mRNA levels were also markedly reduced after IL-1β/LPS treatment at both time points (Figure 5B; 0.99 in the control vs. 0.29 at day 2, and 0.27 at day 5). The concurrent decrease in the expression of *glul* and *snat3* mRNAs suggests that glutamine synthesis is downregulated, while the intracellular glutamine should accumulate due to the repression of its transporter. In agreement with this hypothesis, we measured a significant increase in the intracellular concentration of glutamine following the IL-1β/LPS-induced stress (Figure 5C; 12.61 μM in control vs. and 38.65 μM after 5 days of treatment).

Together, these findings indicate that inflammatory stress evokes an adaptation of the glutamate-glutamine cycle in astrocytes. This adaptation leads to the intracellular retention of both glutamate and glutamine, suggesting a possible change in the neuron–astrocyte metabolic interaction during prolonged exposure to IL-1β/LPS conditions.

Another important role of astrocytes is to provide energy to nearby neurons in the form of lactate [19,20]. To explore how inflammatory stress impacts this metabolic support, we analyzed the expression of key genes involved in lactate production and export. First, we checked whether glucose uptake, which is used to produce lactate, is affected by stress. The mRNA levels for the primary glucose transporter, *glut1*, remained unchanged after incubation with IL-1β/LPS, with a value of 0.93 in controls vs. 1.07 on day 5 (Figure 6A).

We next investigated the mRNA expression of *ucp4*, a mitochondrial uncoupling protein that regulates membrane potential during oxidative phosphorylation and prevents excessive reactive oxygen species (ROS) production. *Ucp4* transcript levels remained unchanged in response to the pro-inflammatory treatment (Figure 6B), indicating that mitochondrial respiration is not markedly affected. In contrast, the transcripts encoding the hexokinase 1 and 2 (*hk1* and *hk2*), enzymes that catalyze the first step of glycolysis, were both significantly upregulated after 5 days of IL-1β/LPS treatment (Figure 6C,D; 0.90 in the control vs. 1.52 at day 5 for *hk1* and 1.19 in the control vs. 3.96 at day 5 for *hk2*).

Together, these data indicate that inflammatory stress upregulates the glycolytic activity in astrocytes, while leaving mitochondrial oxidative phosphorylation unchanged. This metabolic shift may thus support increased lactate production to meet the energy needs of neighboring neurons under chronic inflammatory conditions.

Since glycolytic activity seems increased in astrocytes exposed to IL-1β/LPS stress without impacting mitochondrial respiration, we expected a subsequent effect on lactate production. In fact, the transcript levels for the lactate-producing enzyme, lactate dehydrogenase (LDH), were significantly elevated after the IL-1β/LPS treatment (Figure 7A; 0.94 in the control vs. 1.93 at day 2 and 2.53 at day 5 for *ldh*). Correspondingly, intracellular lactate levels also significantly increased after 5 days of IL-1β/LPS treatment (Figure 7B; 12.49 μM in control versus 18.1 μM at day 5).

Considering the increase in lactate production, we next assessed the expression levels of the monocarboxylate transporters *mct1* and *mct4*, which mediate lactate export from astrocytes into the interstitial space [20]. Although they showed distinct expression patterns, the levels of *mct1* and *mct4* transcripts were both upregulated following treatment (Figure 7C,D). Specifically, *mct1* expression increased after 2 days (Figure 7C; 0.96 in the control vs. 2.02) and remained stable at D5 (Figure 7C; 0.96 vs. 1.62), although these changes were not statistically significant. In contrast, *mct4* expression rose rapidly at D2 (1.02 in control vs. 4.52) and continued to increase further at day 5 (Figure 7D; 1.02 in the control D5 vs. 6.44 in D5 treatment), suggesting a more sustained role in lactate export under prolonged stress. Consistent with the transcriptional up-regulation of these transporters, the extracellular concentration of lactate slightly but significantly increased after 5 days of treatment (Figure 7E; 82 μM vs. 89.3 μM). Together, these data reveal that inflammatory stress promotes both lactate production and export by astrocytes, likely to support the energy demands of neurons under these conditions.

Over time, the diversity of astrocytes, their reactivity in pathological contexts, and their ability to adapt to their environment have become key areas of interest. Since cultured astrocytes can adapt to pro-inflammatory stress, we therefore asked whether these adaptations are reversible. We compared the expression levels of selected genes (*eaat2*, *mct4*, *snat3* and *ldh*) in astrocytes exposed for 5 consecutive days to IL-1β/LPS with those maintained for 2 days in IL-1β/LPS, followed by a 3-day recovery period in control medium (Figure 8A). As previously shown *eaat2* transcript levels were barely significantly increased after 5 days of treatment (Figure 8B). However, this elevated *eaat2* expression persisted for 3 days after the stress exposure was stopped, with no return to baseline levels (Figure 8B), suggesting a long-lasting adaptive mechanism. In contrast, *mct4* expression, which increased sharply after 2 and 5 days of IL-1β/LPS treatments (Figure 7D), returned to near basal levels after the recovery period (Figure 8C). Similarly, *snat3* transcript levels, which were dramatically downregulated in inflammatory conditions, partially returned to basal levels upon stress removal (Figure 8D). In addition, the elevated expression of *ldh* after a 5-day incubation with IL-1β/LPS is also significantly reduced, although it remained elevated when compared to untreated control astrocytes (Figure 8E). Altogether, these data reveal that astrocytic responses are bidirectional with transient (e.g., *mct4*) or more persistent (e.g., *eaat2*) transcriptional adaptations to IL-1β/LPS stress.

## 4. Discussion

Together, our findings demonstrate that primary cultured astrocytes autonomously adapt in response to a pro-inflammatory environment. To investigate the molecular basis of this adaptation upon exposure of cells to a mixture of IL-1β and LPS, we performed experiments over several days of treatment. In this regard, most previous works addressing similar questions were achieved over significantly shorter durations [21]. We used primary cultures of isolated astrocytes, maintained in a defined, neuron-free medium to eliminate any influence from neuronal cross-regulation. Therefore, this experimental paradigm allowed us to assess astrocyte-specific adaptive responses independently of neuronal presence.

Our initial experiments revealed that, as previously reported [15], IL-1β alone has minimal effect on murine astrocytes, whereas LPS alone triggers a robust inflammatory response, with the production and secretion of IL-6 and TNF-α (Figure 2). These pro-inflammatory cytokines likely prolong the physiological effects of the IL-1β and LPS pathways. To better understand the interplay between these signaling pathways, we combined IL-1β and LPS in our experimental setup to measure the effects of both TLR4 (LPS) [16], and NF-kB (IL-1β) signaling pathways [17]. Indeed, despite the discreet effect of IL-1β alone on murine astrocytes, previously published data and our current study demonstrate that IL-1β potentiates the astroglial effects of pro-inflammatory molecules [14], thereby justifying our experimental design. Importantly, we first demonstrated that our combined inflammatory treatment did not alter cell viability or membrane integrity, as assessed by ACT and extracellular measurements, respectively. However, although the toxic effects of LPS have been reported [11,22], we were not able to observe similar results. There could be several reasons for this unexpected result. For instance, different serum batches in the same culture media may exhibit distinct properties. Primary cultures are also vulnerable to protocol differences, so we can assume they might possess stress-resistance capabilities.

It is well known that astrocytes play a central role in regulating glutamate homeostasis in the CNS [6]. They express two key transporters, EAAT2 and EAAT1, which are critical for clearing glutamate from the synaptic cleft. Dysregulation of these transporters can lead to excitotoxicity and consequently cause neurological conditions. Although their upregulation can significantly slow disease progression, it rarely alters the clinical outcome [23]. Here, we measured a rapid increase in the *eaat2* mRNA expression levels, which was not reversed after returning to basal control conditions (Figure 4 and Figure 8), revealing long-term adaptive astrocytic responses.

It is important to note that even if intracellular glutamate concentration increased, it could not be consumed via energy production pathways. The mRNA levels of glutamate dehydrogenase *glud*, which converts glutamate into α-ketoglutarate, remained unchanged. However, the mRNA levels of the Gln synthesis enzyme (*glul*) and its astrocyte-specific transporter (*snat3*) were significantly downregulated (Figure 5). Presumably, these downregulations likely limit the supply and export of glutamine required for glutamate synthesis, thereby reducing the substrate availability for de novo neuronal glutamate synthesis. Notably, this mechanism occurs independently of the neuronal presence, suggesting an astrocyte-intrinsic response that restricts excessive glutamate cycling and prevents neurotoxicity.

These results are particularly intriguing, as previous studies have reported increased glutamine synthetase levels in post-ischemic brains [24,25] and elevated glutamine concentrations [26]. Our data instead suggest an alternative adaptive mechanism in which the downregulation of *snat3* transcripts prevents glutamate accumulation and excitotoxicity. This is consistent with prior findings on a model of major depressive disorder (MDD) induction, showing reduced SNAT3 expression in the striatum and hippocampus 24 h after LPS-induced inflammation [27], although our extended exposure model (5 days) likely accounts for the distinct kinetics and regulation observed here. Overall, our data support the notion of a highly autonomous and regulated glutamate–glutamine cycle in astrocytes under inflammatory stress.

We also analyzed the impact of IL-1β/LPS stress on astrocyte-mediated energy supply. Astrocytes play a crucial role in neuronal support, mainly by producing and exporting lactate [28]. Under our experimental conditions, inflammation strongly affected the lactate production pathway. Although glucose input did not change, we found that genes involved in glycolysis (*hk1*, *hk2*, and *ldh*) were upregulated (Figure 6), leading to elevated intracellular lactate levels. In addition, enhanced expression of the two monocarboxylate transporters, MCT1 and MCT4 (Figure 7), enables lactate export to the extracellular environment. This indicates a metabolic shift toward aerobic glycolysis, a Warburg-like phenotype, in which ATP production increasingly relies on glycolytic flux even under normoxic conditions. Our findings extend previous reports showing upregulation of MCT1 and MCT4 after 24 h of LPS exposure [27]. This is also in line with earlier reports showing an increase in MCT4 under ischemic conditions [28] and its contribution to neuronal survival in co-cultures [29]. Conversely, reduced *mct4* or *eaat1* expression levels resulted in neuronal hyperexcitability [30], further supporting the protective role of astrocyte metabolic support in neuronal energy supply under stress conditions. This shift likely reflects a metabolic adaptation to maintain energy homeostasis and to meet the increased bioenergetic demands associated with the inflammatory state. Similar glycolytic reprogramming has been described in reactive astrocytes, where lactate produced by astrocytes is exported to neurons as an alternative energy source, a process known as the astrocyte–neuron lactate shuttle [31]. Consistent with this, *mct1* and particularly *mct4* expression were upregulated, supporting increased lactate export capacity. The substantial induction of *mct4 MCT4* after 5 days indicates sustained adaptation, as MCT4 is typically associated with high-glycolytic astrocytes that supply metabolic support to neurons. Therefore, inflammatory stress seems to induce a metabolically protective phenotype, characterized by elevated glycolysis and lactate release [32].

Finally, our study demonstrates that several of these inflammation-induced adaptations are reversible. While *eaat2* expression remained elevated even after removal of inflammatory stimuli, *mct4* and *snat3* expression levels returned to baseline, and *ldh* expression partially normalized (Figure 8). This suggests that astrocytes retain a degree of plasticity, allowing them to adjust their function in response to environmental cues dynamically, and indicates that metabolic reprogramming is more dynamic and reversible. This dichotomy between stable (e.g., EAAT2) and reversible (e.g., MCT4) responses highlights the selective plasticity of astrocytes under chronic stress. Astrocytes may maintain enhanced glutamate clearance as a long-term protective mechanism while flexibly adjusting their metabolic outputs in response to changing environmental conditions. This dual nature reflects the growing understanding that astrocyte reactivity is not binary but varies from neuroprotective (A2) to neurotoxic (A1) and rather exists along a continuum of functional states [33].

Our work aimed to unravel the specific adaptive response of astrocytes in the complex environment of neuroinflammation onset and subsequent effects. Indeed, the pathophysiological context of neuroinflammation is a highly dynamic system, driven by multifactorial mechanisms and characterized by tight intercellular crosstalk and adjustments. To restrict our investigations to astrocyte behavior, we used pure isolated primary astrocytes treated with an IL-1β/LPS mixture over several days. Although it is challenging to mimic deleterious chronic conditions observed in vivo using in vitro experiments, IL-1β and LPS are two well-known neuroinflammation mediators used both in vivo and in vitro. Overall, even if this model is not optimal, our results allow us to visualize the molecular mechanisms of adaptation in vitro. Despite its limitations, we observe adaptations consistent with recent findings. For example, most neurodegenerative diseases, such as Alzheimer’s, Parkinson’s, and Huntington’s, exhibit impaired glutamate uptake [30] and, consistent with this, our results show that under IL-1β/LPS stress, astrocytes engage coordinated responses to enhance glutamate uptake and limit the glutamine export. This serves as a backup mechanism for neighboring neurons, providing neuroprotective effects.

Chronic inflammatory stress triggers a complex yet coordinated reprogramming of astrocyte function. Astrocytes remain viable and reactive, increasing EAAT2-mediated glutamate uptake, suppressing the glutamate–glutamine cycle, and enhancing glycolytic metabolism and lactate export. Some of these changes, particularly EAAT2 upregulation, persist after stress ends, indicating lasting transcriptional imprinting, while others are reversible, demonstrating metabolic flexibility. Overall, these findings emphasize the crucial role of astrocyte metabolic plasticity in maintaining neuronal homeostasis during inflammation and suggest that prolonged or abnormal reactivity could contribute to neuroinflammatory and metabolic diseases. Although the in vitro model we used is a powerful tool for studying these plastic processes and could also serve as a valuable platform for drug screening, it is essential to examine its relevance to CNS function, particularly regarding astrocyte–neuron communication. Therefore, the effect of stress-preconditioned astrocytes on neuronal phenotype in co-cultures needs further investigation. Ultimately, we should be able to evaluate the therapeutic potential of drugs that enhance astrocyte neuroprotection, an emerging and promising area of research [33].

## Figures and Tables

**Figure 1 cells-14-01737-f001:**
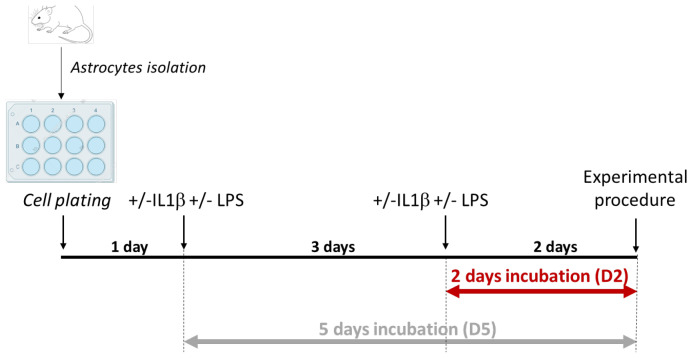
Experimental design. Outline of the experimental protocol and pro-inflammatory treatments used on primary cultured astrocytes.

**Figure 2 cells-14-01737-f002:**
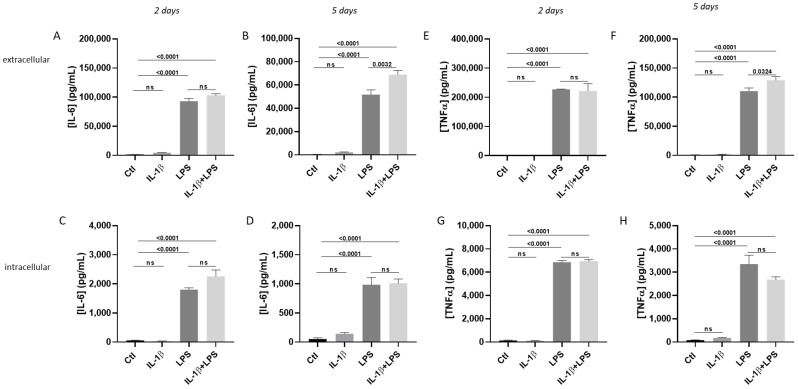
Levels of secreted (**A**,**B**,**E**,**F**) and intracellular (**C**,**D**,**G**,**H**) cytokines produced by astrocytes in response to IL-1β/LPS treatments. IL-6 (**A**–**D**) and TNF-α (**E**–**H**) concentrations were measured in astrocyte primary cell cultures incubated with IL-1β, LPS, or a combination of both, and compared with untreated control cells (Ctl). Data are presented as the representative (out of 3 experiments), mean ± SEM of 3 independent replicates. Indicates statistical differences analyzed by ordinary one-way ANOVA with a Tukey post hoc test for multiple comparison data sets and a Ratio paired *t*-test; *p*-values are shown on the bars. ns: non-significant.

**Figure 3 cells-14-01737-f003:**
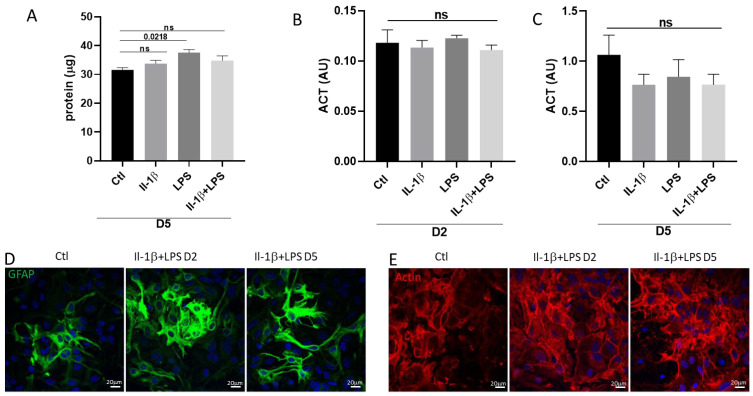
Astrocyte survival and morphology in response to IL-1β/LPS treatments. (**A**) Quantification of protein levels in astrocytes cultured for 5 days in the absence or presence of IL-1β, LPS, or a combination of both. No significant decrease in protein concentration (ns) between conditions was measured. (**B**,**C**) ACT measurements for cell viability of cultured astrocytes for 2 (**B**) and 5 days (**C**) under the indicated treatments. (**D**) Immunofluorescence of GFAP on cultured astrocytes treated or not with IL-1β/LPS for the indicated time. (**E**) Alexa647-phalloidin labeled polymerized actin in IL-1β/LPS-treated and control astrocytes. Scale bar, 20 μm. (**A**–**C**) Representative data from 3 experiments are presented, mean ± SEM. Ordinary one-way ANOVA with a Tukey post hoc test for multiple comparison data sets and a Ratio paired *t*-test; *p*-values are shown on the bars. ns: non-significant.

**Figure 4 cells-14-01737-f004:**
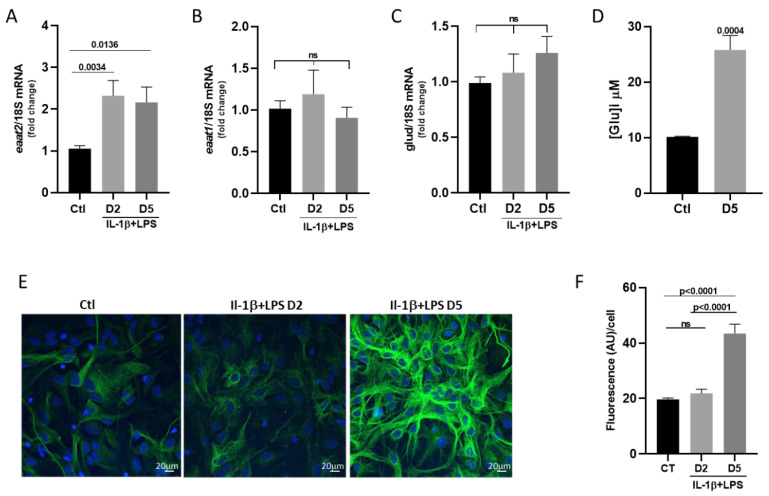
Expression of genes encoding proteins involved in the control of glutamate homeostasis. Relative quantification by RT-qPCR of *eaat2* (**A**), *eaat1* (**B**), *glud* (**C**) transcripts normalized to *18S* ribosomal subunit transcripts in astrocytes cultured with the indicated treatments. (**D**) Intracellular levels of glutamate in astrocyte cultures maintained for 5 days with or without IL-1β/LPS and their corresponding control. Significance by one-sample *t*-test: *p* < 0.001. Error bars indicate sem. (**E**) EAAT2-stained astrocytes after 2 or 5 days of IL-1β/LPS treatment. Representative images are shown. Scale bars, 20 μm. (**F**) Levels of EAAT2 fluorescence quantified on images prepared as in (**E**). (**A**–**C**) Data presented are the mean ± SEM of four independent experiments. Statistical significance is analyzed by ordinary one-way ANOVA with a Tukey post hoc test for multiple comparison data sets and a Ratio paired *t*-test; *p*-values are indicated on the bars. ns: non-significant.

**Figure 5 cells-14-01737-f005:**
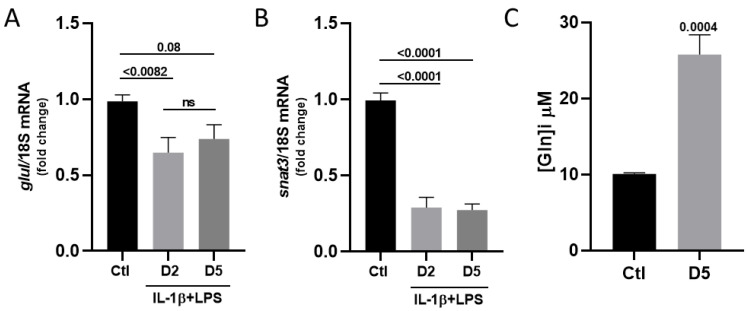
Astrocytic regulation of the glutamate-glutamine cycle in response to IL-1β/LPS stress. Relative quantification of *glul* (**A**) and *snat3* (**B**) transcripts by RT-qPCR. Astrocytes were treated with IL-1β/LPS for 2 or 5 days (D2 and D5), the expression level was compared with the “non-treated” control (Ctl), and normalized to the levels of the 18S ribosomal subunit transcripts. (**A**,**B**) Data are presented as the mean± SEM of 5 independent experiments. Statistical significance is analyzed by ordinary one-way ANOVA with a Tukey post hoc test for multiple comparison data sets and a Ratio paired *t*-test; *p*-values are indicated on the bars. ns: non-significant. (**C**) Intracellular levels of glutamine in cultured astrocytes maintained for 5 days with IL-1β/LPS and their corresponding control. Significance by one-sample *t*-test. Data are presented as means ± SEM of 3 independent experiments.

**Figure 6 cells-14-01737-f006:**
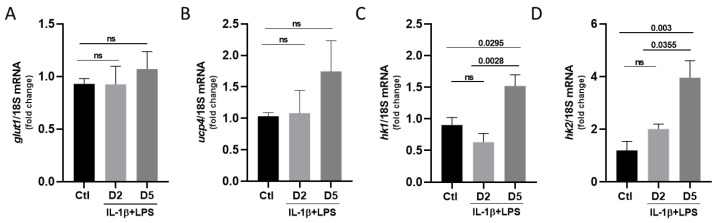
Exposure of astrocytes to IL-1β/LPS alters cellular pathways regulating glucose metabolism. Relative quantification of the *glut1* (**A**), *ucp4* (**B**), *hk1* (**C**), and *hk2* (**D**) transcripts by RT-qPCR. Astrocytes were treated with IL-1β/LPS for 2 or 5 days (D2 and D5) and the expression level was compared with the “non-treated” control (Ctl) and normalized to the levels of the 18S ribosomal subunit transcript. Data are presented as mean ± SEM of 5 independent experiments. Statistical significance is analyzed using ordinary one-way ANOVA with a Tukey post hoc test for multiple comparison data sets and a Ratio paired *t*-test; *p*-values are indicated on the bars. ns: non-significant.

**Figure 7 cells-14-01737-f007:**
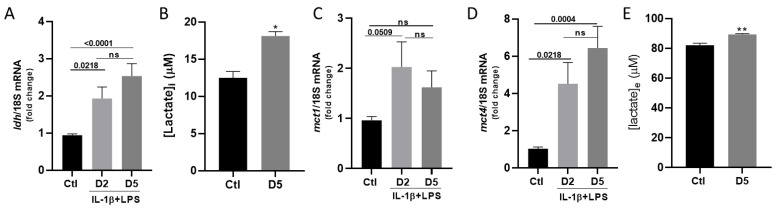
Cultured astrocytes upregulate lactate production and release in response to IL-1β/LPS treatments. (**A**) RT-qPCR analysis of the *ldh* transcripts relative to the 18S ribosomal subunit transcripts after treatment with IL-1β/LPS or control. (**B**) Intracellular levels of lactate in cultures of astrocytes maintained for 2 or 5 days with IL-1β/LPS or the corresponding control. (**C**,**D**) Relative quantitation of the *mct1* (**C**) and *mct4* (**D**) transcripts by RT-qPCR relative to the levels of the 18S ribosomal subunit transcripts upon the indicated treatments. (**E**) Extracellular levels of lactate from cultured astrocyte medium were maintained for 2 or 5 days with IL-1β/LPS or the corresponding control. Data are presented as the mean ± SEM of 5 experiments. Statistical significance is analyzed by ordinary one-way ANOVA with a Tukey post hoc test for multiple comparison data sets and a Ratio paired *t*-test; *p*-values are indicated on the bars. ns: non-significant. (**B**,**E**) Significance by one-sample *t*-test: * *p* < 0.05, ** *p* < 0.01. Data are presented as the mean ± SEM of 3 independent experiments.

**Figure 8 cells-14-01737-f008:**
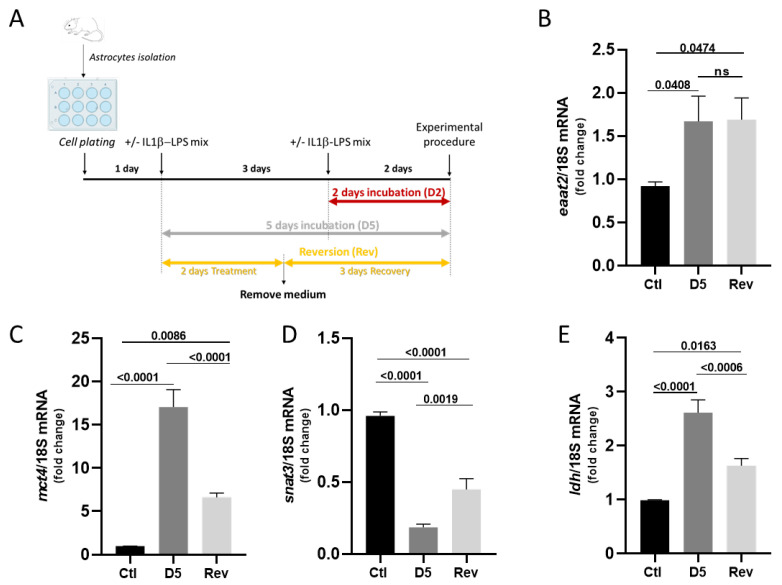
Reversibility of adaptive astrocytic mechanisms after stopping exposure to IL-1β/LPS treatments. (**A**) Experimental protocol for testing the recovery of primary astrocyte cultures after exposure to IL-1β and LPS. (**A**–**E**) Relative quantification by RT-qPCR of the *eaat2* (**B**), *mct4* (**C**), *snat3* (**D**), and *ldh* (**E**) transcripts relative to the levels of the 18S ribosomal subunit transcript in astrocytes cultured in the presence of IL-1β/LPS for 2 days and then returned to control media for 3 days or maintained for 5 days with IL-1β/LPS. Data are shown as means ± SEM of 4 separate experiments. Statistical significance was assessed using an ordinary one-way ANOVA with a Tukey post hoc test for multiple comparison data sets and a Ratio paired *t*-test; *p*-values are indicated on the bars. ns: non-significant.

## Data Availability

The datasets generated and analyzed during the current study are available from the corresponding authors on reasonable request.

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
