# Peer review of "Adaptive Gene Expression Induced by a Combination of IL-1β and LPS in Primary Cultures of Mouse Astrocytes"

_cells, 2025, doi:10.3390/cells14211737_

Round 1

Reviewer 1 Report

Comments and Suggestions for Authors

The manuscript entitled “A model of astrocytes primary culture for physiological adaptative studies” addresses an important and timely topic and presents a study of considerable scientific interest. The results obtained are numerous and provide a broad and valuable overview of astrocytic responses under the experimental conditions. The authors convincingly demonstrate, in an in vitro model, certain adaptive properties of glial cells when exposed to interleukin-1β and LPS. However, despite the scientific merit of the findings, the manuscript itself is written in a rather disorganized manner and requires further refinement to improve clarity and coherence.

My comments are as follows:

  1. The title of the manuscript does not fully reflect its content. The study includes not only factors that could be considered at the borderline of physiology. Moreover, the title does not clearly indicate the specific experimental settings under which astrocytes were examined in order to reveal their adaptive responses.
  2. The introduction does not sufficiently introduce the reader to the subject or to the aim of the study. Although the authors state that they investigate the “long-term consequences of lipopolysaccharide (LPS) and interleukin-1” as pro-inflammatory factors, the introduction lacks a description that would connect this aim with the expected cellular responses under inflammatory conditions. Furthermore, there is no reference to how astrocytes behave under the influence of LPS and IL-1 based on previous studies. The authors also do not outline which specific factors (proteins) are examined and why these particular molecules were selected, nor do they provide a rationale grounded in the literature. Instead, the introduction mainly consists of general information about certain astrocytic functions, including their role in neurotransmission and the importance of transporters in these processes. The introduction should shed a light on the studies that are demonstrated in the article.
  3. In the introduction, the Authors point the importance of obesity, diabetes and high-fat food intake in the processes affecting cognition and vulnerability to neurodegenerative diseases without any reflection in the discussion section or conclusions.
  4. In the introduction, the Authors mention that some issues are “well documented” (e.g. line 40, 49) giving only one reference. Same in the discussion, the Authors use phrases like: “most previous works” (line 377) or “previous reports showing…” (line 426) also giving only one reference. So, where there more reports, references or only one in each case?
  5. The Authors studied the influence of LPS and IL-1 or both on the astrocytes for 2 and 5 days. Is a 5-day period sufficient to be considered a long-term exposure?
  6. What is the hypothesis for the posed research question? It is difficult to assess the purpose of the study without a proper introduction. Without this, it is difficult to make any assumptions regarding the results.
  7. In Materials and methods it was not stated how many C57BL/6 female mice were taken for the experiment and what was the age of the animals. The Authors state that neonatal brains (P0-P3) were dissected for the studies. Were they purchased as neonates or there were pregnant dams?
  8. The Authors do not mention any Ethics Committee Agreement for the studies and procedures conducted on animals. They mention only the directives and guidelines for animal treatment.
  9. The experimental design is unclear. The text should provide a detailed description of the sequential procedures that were performed. The materials and methods do not give information about the control and experimental groups. Figure 1 is not entirely understandable, and it lacks a proper legend. It is unclear what D2 and D5 refer to. The method of administration of IL-1 and LPS is not specified—whether they were given together or separately. The text refers to days, while the figure shows hours. Much has to be inferred from the text, whereas it should be clearly described in the Materials and Methods section. The figure illustrating the experimental design should be self-explanatory.
  10. It is unclear, why the combination of IL1+LPS was implied for the study.
  11. In many places of the manuscript, there are double spaces, and the beta symbol is missing for interleukin, as well as the alpha symbol for TNF.
  12. The Authors should check if the abbreviations that are used in the text are explained in the place where they appear for the first time, e.g. “tumor necrosis factor” and other… some are not explained in the text.
  13. The method of total immunolabeling from immunostaining with use of ImageJ software should be shortly described or indicated in the literature.
  14. Are the Authors sure that they used Standard error of the mean and not a standard deviation?
  15. Homogeneity of variance is a key assumption for ANOVA, as the test relies on the idea that the compared groups have similar variability. If this assumption is violated, the results of ANOVA may become unreliable, leading to an increased risk of Type I or Type II errors. Therefore, testing for homogeneity of variance is essential to ensure the validity of the statistical conclusions. The Authors state that only a normal distribution was tested.
  16. “*p < 0.05 was considered significant” is a shorthand expression and not a properly constructed sentence. Why the Authors use “*p” symbol instead of “p”? Same for figure 2 (line 185), there is “*” as the indicator of significance, but it was not used on the figure.
  17. In the Results section, there are some paragraphs that are not presenting a strict result, e.g. lines 166-168, 175-178, 251-253, 267-270, 308-315… and other similar. Some of the provided information should have been presented at the planning stage as an introduction to the topic and to explain the rationale for the chosen actions and experiments. Other parts should be placed in the discussion section.
  18. In Figure 3, there is no explanation for the arrowheads on the photomicrographs. They are mentioned in the text but not in the legend of the figure.
  19. Line 235. Did Authors study correlations?
  20. In the legend of figure 5 and 6 the statements “…presence or absence of IL-1 LPS for 2 or 5 days…” and “…presence or absence of IL-1   LPS for various durations…” are unclear.
  21. I am not convinced that the term “inflammation conditions” is used appropriately in this study. Brain inflammation induced by LPS involves not only the response of astrocytes, but also other brain cells such as neurons and microglia, as well as peripheral immune cells and endothelial cells. All these cell types produce a wide array of cytokines and chemokines, leading to complex interactions between them. Therefore, it might be more accurate to state that, in this study, the cells were simply exposed to IL-1 and/or LPS, rather than to inflammatory conditions in general.
  22. Are the Authors aware of any limitations to their studies? The limitations are not mentioned in the discussion section.
  23. Considering the large number of results obtained by the authors and presented in this manuscript, the conclusions appear rather modest. It would be necessary to clearly present what can be inferred from the obtained results, providing a thorough summary of the study.
  24. It should be noted that the authors did not distinguish whether the astrocytes in their culture represented the A1 or A2 phenotypes. This distinction is important because A1 astrocytes are generally pro-inflammatory and potentially neurotoxic, whereas A2 astrocytes are neuroprotective and supportive of regeneration. In an in vitro culture, without the presence of microglia and other brain cells, it is unclear whether exposure to IL-1β or LPS induces an A1-like or A2-like response. This limitation could affect the interpretation of the observed adaptive responses, and it should be acknowledged in the discussion.

In summary, while the study presents interesting and valuable data, considering all the points raised above, the manuscript requires substantial improvement. It could be considered for publication after a major revision.

Comments on the Quality of English Language

The English language in the manuscript should be improved. In some phrases, the authors use American English rather than British English. Moreover, they often employ shorthand expressions that are not complete, logically structured sentences. Such expressions may be acceptable in figure legends, but not in the main text e.g. "*p < 0.05 was considered significant."

Author Response

The manuscript entitled “A model of astrocytes primary culture for physiological adaptative studies” addresses an important and timely topic and presents a study of considerable scientific interest. The results obtained are numerous and provide a broad and valuable overview of astrocytic responses under the experimental conditions. The authors convincingly demonstrate, in an in vitro model, certain adaptive properties of glial cells when exposed to interleukin-1β and LPS. However, despite the scientific merit of the findings, the manuscript itself is written in a rather disorganized manner and requires further refinement to improve clarity and coherence.

Thank you for the kind comments and suggestions.

My comments are as follows:

  1. The title of the manuscript does not fully reflect its content. The study includes not only factors that could be considered at the borderline of physiology. Moreover, the title does not clearly indicate the specific experimental settings under which astrocytes were examined in order to reveal their adaptive responses.

Thank you for your valuable suggestion. We have revised the title based on the reviewer’s comment. The new title is: «Adaptive gene expression induced by a combination of IL-1b and LPS in primary cultures of mouse astrocytes »

  1. The introduction does not sufficiently introduce the reader to the subject or to the aim of the study. Although the authors state that they investigate the “long-term consequences of lipopolysaccharide (LPS) and interleukin-1” as pro-inflammatory factors, the introduction lacks a description that would connect this aim with the expected cellular responses under inflammatory conditions. Furthermore, there is no reference to how astrocytes behave under the influence of LPS and IL-1 based on previous studies. The authors also do not outline which specific factors (proteins) are examined and why these particular molecules were selected, nor do they provide a rationale grounded in the literature. Instead, the introduction mainly consists of general information about certain astrocytic functions, including their role in neurotransmission and the importance of transporters in these processes. The introduction should shed a light on the studies that are demonstrated in the article.

 We agree with these remarks, and we have revised the introduction accordingly. Specifically, we mentioned previous studies related to the effects of IL-1β and LPS on astrocytes, and we more carefully introduced the molecular targets studied in the paper (lines 72-98).

  1. In the introduction, the Authors point the importance of obesity, diabetes and high-fat food intake in the processes affecting cognition and vulnerability to neurodegenerative diseases without any reflection in the discussion section or conclusions.

Thank you for your helpful observation. We have toned down this point in the revised manuscript.

  1. In the introduction, the Authors mention that some issues are “well documented” (e.g. line 40, 49) giving only one reference. Same in the discussion, the Authors use phrases like: “most previous works” (line 377) or “previous reports showing…” (line 426) also giving only one reference. So, where there more reports, references or only one in each case?

 We apologize for this point. The text has been modified accordingly (lines 426, 486).

  1. The Authors studied the influence of LPS and IL-1 or both on the astrocytes for 2 and 5 days. Is a 5-day period sufficient to be considered a long-term exposure?

Thank you for your comment. As noted in the introduction, most previous studies showed exposure for up to 2 days or less. Based on our experience, 5 days is the longest we can expose the cells under such conditions without losing them. We believe this is a longer exposure compared to those previously reported.

  1. What is the hypothesis for the posed research question? It is difficult to assess the purpose of the study without a proper introduction. Without this, it is difficult to make any assumptions regarding the results.

The introduction has been mostly reorganized to address your second point. We thank you for your comment, and believe these changes improve the clarity of our study.

  1. In Materials and Methods, it was not stated how many C57BL/6 female mice were taken for the experiment and what was the age of the animals. The Authors state that neonatal brains (P0-P3) were dissected for the studies. Were they purchased as neonates or there were pregnant dams?

As mentioned in the Materials and Methods section, the primary cultures were performed using newborn pups from in-house-bred pregnant mice aged 10 to 20 weeks old.

  1. The Authors do not mention any Ethics Committee Agreement for the studies and procedures conducted on animals. They mention only the directives and guidelines for animal treatment.

Since our experimental procedures (performed using newborn pups) do not require authorization for animal experimentation there is no need to mention any Ethics Committee Agreement.

  1. The experimental design is unclear. The text should provide a detailed description of the sequential procedures that were performed. The materials and methods do not give information about the control and experimental groups. Figure 1 is not entirely understandable, and it lacks a proper legend. It is unclear what D2 and D5 refer to.

We thank you for your valuable suggestion, and we have now modified the figure 1 and made the requested changes. D2 and D5 were exchanged by « 2 days incubation (D2) » and « 5 days incubation (D5) », respectively.

  1. The method of administration of IL-1 and LPS is not specified—whether they were given together or separately. The text refers to days, while the figure shows hours. Much has to be inferred from the text, whereas it should be clearly described in the Materials and Methods section. The figure illustrating the experimental design should be self-explanatory.

Thank you for your remarks. Figure 1 summarizes the experiments using IL-1b treatment or LPS treatment separately or together. Thus, we kept the following nomenclature in Figure 1: +/-IL-1b+/-LPS. On the contrary, Figure 8 resumes the treatment and analysis of astrocytes using a mixture of IL-1β and LPS. Changes have been made as follows: +/-IL-1b+/-LPS è +/-IL-1b-LPS mix.

  1. It is unclear, why the combination of IL1+LPS was implied for the study.

We agree with this observation and have added further arguments in the introduction to clarify it (Line 98-105).

  1. In many places of the manuscript, there are double spaces, and the beta symbol is missing for interleukin, as well as the alpha symbol for TNF.

Sorry for this recurring mistake caused by the formatting process during submission. Corrections have been made in the revised manuscript.

  1. The Authors should check if the abbreviations that are used in the text are explained in the place where they appear for the first time, e.g. “tumor necrosis factor” and other… some are not explained in the text.

Thank you for your comment. The abbreviation list has been updated, and each abbreviation is now explained clearly the first time it appears in the text.

  1. The method of total immunolabeling from immunostaining with use of ImageJ software should be shortly described or indicated in the literature.

Thank you for your suggestion. A relevant reference has been added (Schneider, C.A.; Rasband, W.S.; Eliceiri, K.W. NIH Image to ImageJ: 25 years of image analysis. Nat Methods 2012, 9, 671-675, doi:10.1038/nmeth.2089).

  1. Are the Authors sure that they used Standard error of the mean and not a standard deviation?

All the data shown are expressed as the mean ± SEM. This is now clearly stated in the revised manuscript.

  1. Homogeneity of variance is a key assumption for ANOVA, as the test relies on the idea that the compared groups have similar variability. If this assumption is violated, the results of ANOVA may become unreliable, leading to an increased risk of Type I or Type II errors. Therefore, testing for homogeneity of variance is essential to ensure the validity of the statistical conclusions. The Authors state that only a normal distribution was tested.

We completely agree with this critical point concerning the reliability of statistical analysis. We confirmed that, for each ANOVA test, the homogeneity of variance (variance between groups is approximately equal) has been verified, as the F value is consistently well above 1. This has been added to the « Materials and Methods » section. In addition, since the one-way ANOVA does not assume a strict normal distribution (an approximately normal distribution is tolerated), we removed the following sentence from the text: “All data were tested for normal distribution.”

  1. “*p < 0.05 was considered significant” is a shorthand expression and not a properly constructed sentence. Why the Authors use “*p” symbol instead of “p”? Same for figure 2 (line 185), there is “*” as the indicator of significance, but it was not used on the figure.

Thank you for your valuable observation. We have made all the requested changes in the revised manuscript (line 235, 261, 296, 324, 357, 390).

  1. In the Results section, there are some paragraphs that are not presenting a strict result, e.g. lines 166-168, 175-178, 251-253, 267-270, 308-315… and other similar. Some of the provided information should have been presented at the planning stage as an introduction to the topic and to explain the rationale for the chosen actions and experiments. Other parts should be placed in the discussion section.

We agree that the sections cited do not strictly describe the results obtained, but rather serve to introduce or explain them. The additional details provided in the new version of the manuscript in the « introduction » and « discussion » sections could allow us to remove this information from the « results » section. However, we believe that it does not interfere with the comprehensive reading of the manuscript and may help to better contextualize and appreciate the results. Therefore, no changes have been made to these sections.

  1. In Figure 3, there is no explanation for the arrowheads on the photomicrographs. They are mentioned in the text but not in the legend of the figure.

Since these arrowheads were not helpful for a better understanding, we have suppressed them.

  1. Line 235. Did Authors study correlations?

No correlation between glutamate accumulation and EAAT2 upregulation has been calculated. Therefore, the term « correlates » was removed from the revised text (line…).

  1. In the legend of figure 5 and 6 the statements “…presence or absence of IL-1 LPS for 2 or 5 days…” and “…presence or absence of IL-1   LPS for various durations…” are unclear.

The text has been amended as requested (lines 320 and 352).

  1. I am not convinced that the term “inflammation conditions” is used appropriately in this study. Brain inflammation induced by LPS involves not only the response of astrocytes, but also other brain cells such as neurons and microglia, as well as peripheral immune cells and endothelial cells. All these cell types produce a wide array of cytokines and chemokines, leading to complex interactions between them. Therefore, it might be more accurate to state that, in this study, the cells were simply exposed to IL-1 and/or LPS, rather than to inflammatory conditions in general.

Thank you for your valuable suggestion. We agree that IL-β and LPS treatments, even when administered simultaneously, do not fully restore an inflammatory state. Although they are both well documented as priming or at least essential factors in the onset or maintenance of neuroinflammation, we understand that the frequent use of « inflammation conditions » in the first version of the manuscript can be confusing. Therefore, we have mostly, but not systematically, replaced « inflammation condition » with « IL-b/LPS treatment » especially in the « results » section.

  1. Are the Authors aware of any limitations to their studies? The limitations are not mentioned in the discussion section.

We thank you for this comment. The limitations of the study have been included in the revised discussion section.

  1. Considering the large number of results obtained by the authors and presented in this manuscript, the conclusions appear rather modest. It would be necessary to clearly present what can be inferred from the obtained results, providing a thorough summary of the study.

Thank you for your valuable suggestion. The conclusions were expanded consistently with our objectives and the current state of the art.

  1. It should be noted that the authors did not distinguish whether the astrocytes in their culture represented the A1 or A2 phenotypes. This distinction is important because A1 astrocytes are generally pro-inflammatory and potentially neurotoxic, whereas A2 astrocytes are neuroprotective and supportive of regeneration. In an in vitro culture, without the presence of microglia and other brain cells, it is unclear whether exposure to IL-1β or LPS induces an A1-like or A2-like response. This limitation could affect the interpretation of the observed adaptive responses, and it should be acknowledged in the discussion.

We agree with this constructive observation and apologize for overlooking this point. The concepts of A1- and A2- responses were introduced in the discussion section of the manuscript (line 506-513).

In summary, while the study presents interesting and valuable data, considering all the points raised above, the manuscript requires substantial improvement. It could be considered for publication after a major revision.

We appreciate your careful and constructive review of the manuscript and hope the revised version meets your approval for publication.

Reviewer 2 Report

Comments and Suggestions for Authors

The authors conducted an in vitro study on primary cultures of rat astrocytes to examine specific responses to chronic inflammatory stress, induced by prolonged administration of interleukin-1 and LPS. The manuscript is well-written and sufficiently scientifically sound. However, the experimental design is quite basic, and the results lack novelty, appearing somewhat anticipated by current literature on the topic.

Major criticisms are the following:

The Authors claim the relevance of primary astrocytic cultures as a model to understand specific astrocytic adaptive mechanisms occurring during chronic inflammatory challenge. This point raises some concerns. Indeed, the pathways explored by the Authors are all involved in the neuron-astrocytic crosstalk. Moreover, in vivo, astrocytes' metabolic and functional changes upon inflammatory challenge are tightly related to neurons. It is difficult, therefore, to understand the translational value of a “neuron-independent” primary astrocytic culture in this specific context. Perhaps this point deserves a more exhaustive explanation.

The abstract is very generic and should be rewritten, including a short summary of the most relevant results obtained in the study.

In the introduction, the Authors mention the relation between obesity, diabetes and chronic inflammation. However, how primary astroglia cultures treated with LPS and IL-1 may be representative of that specific situation remains elusive.

Most of the data are related to changes in gene expression (qPCR analysis), which does not allow us to conclude that the reported differences in the mRNA expression will also result in different protein levels. A validation at the protein level should be performed.

In the discussion, the Authors state that they “first demonstrated that the combined inflammatory treatment did not alter cell viability”. This result is not in line with other studies, and this discrepancy should be more extensively discussed. In addition, in the discussion section, the novelty and importance of the present results, as well as the extra value provided by their experimental approach, should be highlighted.

Author Response

The Authors claim the relevance of primary astrocytic cultures as a model to understand specific astrocytic adaptive mechanisms occurring during chronic inflammatory challenge. This point raises some concerns. Indeed, the pathways explored by the Authors are all involved in the neuron-astrocytic crosstalk. Moreover, in vivo, astrocytes' metabolic and functional changes upon inflammatory challenge are tightly related to neurons. It is difficult, therefore, to understand the translational value of a “neuron-independent” primary astrocytic culture in this specific context. Perhaps this point deserves a more exhaustive explanation.

Thank you for your valuable observation. As discussed in the relevant section of the revised manuscript, we are aware of the limitations of our study (lines 523-525). However, the results presented in the current manuscript allowed us to envision further experiments to achieve a more physiological context. We are currently developing an experimental approach that will enable to work with co-cultured neuronal and astrocytic cells from different genetic backgrounds. Our ultimate goal is to quantify the impact of preconditioning astrocyte cultures (LPS, hypoxia, metabolic starvation) on cortical neuron phenotype.

The abstract is very generic and should be rewritten, including a short summary of the most relevant results obtained in the study.

We totally agree with this constructive comment. The abstract has been thoroughly rewritten to be less generic and better reflect the goal and results of our study.

In the introduction, the Authors mention the relation between obesity, diabetes and chronic inflammation. However, how primary astroglia cultures treated with LPS and IL-1 may be representative of that specific situation remains elusive.

Thank you for highlighting this aspect as an elusive point. Since this question distracts the reader from a straightforward line of reasoning, we prefer not to use this physiological context to justify our work.

Most of the data are related to changes in gene expression (qPCR analysis), which does not allow us to conclude that the reported differences in the mRNA expression will also result in different protein levels. A validation at the protein level should be performed.

We thank you for this remark and agree with this essential aspect of biology, in which the function of a gene is the protein it encodes. We chose not to experimentally quantify the proteins identified as crucial, because the normalization conditions are too random, depending on the quality of the antibodies and the robustness of the normalization method. We have made several attempts in this direction. Still, as the objective of the work is to demonstrate the adaptability of our cellular model, we do believe that mRNA quantification is sufficient. As indicated in a new title, we focused our work on « gene expression ».

In the discussion, the Authors state that they “first demonstrated that the combined inflammatory treatment did not alter cell viability”. This result is not in line with other studies, and this discrepancy should be more extensively discussed.

We agree that discrepancies exist between our findings and some previously published data on the effects of LPS on astrocyte viability. We believe that differences may result from the variability of the model used, as discussed in the revised discussion section (line 443-447).

In addition, in the discussion section, the novelty and importance of the present results, as well as the extra value provided by their experimental approach, should be highlighted.

Thank you for your helpful suggestion. We have made several changes to the revised discussion section.

We thank you for your clear and constructive review of the manuscript. We hope that you will be satisfied by this revised version.

Reviewer 3 Report

Comments and Suggestions for Authors

The manuscript by Coppola et al describes the change of glial gene transcripts after administration of LPS and IL1beta. The experiments done for preparation of the manuscript are comprehensive. The style of the manuscript has to be considerably improved.

Major points:

  1. In the manuscript are missing α and β. Please write at first the full name of substances and the abbreviation in parentheses. There are lot of abbreviation which are not explained in the text and not mentioned in the list. For example, INT, ACT, GluD. Please arrange the list of abbreviations alphabetical.
  2. The title is misleading. Please specify that you investigate changes in astroglial protein and gene expression after induction if inflammation.
  3. Please check carefully If you indicate in all captions the number of characterized samples and the indication was bar graphs represented (for example mean +/- SEM or SD).
  4. Figure 3,4: Immunohistochemistry. For your experiments you need 2 independent controls, one for D2 and one for D5 with identical duration of culture w/wo cytokines/toxins. Fig. 4: You did not quantify the staining, intensity/area. For what do the arrows in figure 3 stand for.
  5. You find a lot of different and opposed modulations. May you not summarize the different effects on transporters, nutrients in a graphical abstract?

Minor points:

Line 56: … LPS…, two inflammatory markers elevated under diabetic conditions? LPS is not an inflammatory marker but a bacterial endotoxin. It was reported that an LPS-binding protein was increased in adipose tissue.

Line 95: what is INT?

Line 132: Which reference genes were used.

Line 176: Which stress pathways, please introduce the pathways already here.

Author Response

(The authors gave the same response as above.)

Round 2

Reviewer 1 Report

Comments and Suggestions for Authors

The authors have met my expectations regarding the revision of the manuscript and have appropriately addressed my comments. However, I have still some concerns about the manuscript.

I have certain concerns, as the procedures performed on animals — including their euthanasia, which enabled the collection of brains — require approval from an ethics committee. Although the authors conducted experiments on cell cultures, the source of these cells was living animals. Ethical approval is mandatory regardless of the age of the animals, particularly when euthanasia is involved.

Still, the method of total immunolabeling from immunostaining with use of ImageJ software is not described and is not indicated in the literature. The article proposed by the Authors shows a history and overal options that are available in the program. There is no description of the method used.

Schneider, C.A.; Rasband, W.S.; Eliceiri, K.W. NIH Image to ImageJ: 25 years of image analysis. Nat Methods 2012, 9, 671-619 675, doi:10.1038/nmeth.2089.

In my opinion, the manuscript can be published in cells after minor revision.

Author Response

The authors have met my expectations regarding the revision of the manuscript and have appropriately addressed my comments. However, I have still some concerns about the manuscript.

I have certain concerns, as the procedures performed on animals — including their euthanasia, which enabled the collection of brains — require approval from an ethics committee. Although the authors conducted experiments on cell cultures, the source of these cells was living animals. Ethical approval is mandatory regardless of the age of the animals, particularly when euthanasia is involved.

All the requested information has been inserted in the text (lines 113-120), and we have completed the Institutional Review Board Statement: The animal study protocol was approved by the National Animal Care and Ethics Committee (project reference APAFIS #18648-201901111154666 v6, authorization valid until December 10th, 2028). Line 555-557.

Still, the method of total immunolabeling from immunostaining with use of ImageJ software is not described and is not indicated in the literature. The article proposed by the Authors shows a history and overal options that are available in the program. There is no description of the method used.

Schneider, C.A.; Rasband, W.S.; Eliceiri, K.W. NIH Image to ImageJ: 25 years of image analysis. Nat Methods 2012, 9, 671-619 675, doi:10.1038/nmeth.2089.

We have updated the reference article and added more details about the method used. The revised reference is: Abramoff, M.D., Magalhaes, P.J., Ram, S.J. "Image Processing with ImageJ". Biophotonics International, volume 11, issue 7, pp. 36-42, 2004. It can be downloaded here: https://imagej.net/ij/docs/faqs.html.

In my opinion, the manuscript can be published in cells after minor revision.

We thank you for the extensive review work.

Reviewer 2 Report

Comments and Suggestions for Authors

The Authors addressed all the concerns raised by the reviewer and the clarity of the manuscript is improved.

Author Response

The Authors addressed all the concerns raised by the reviewer and the clarity of the manuscript is improved.

We thank you for a very constructive reviewing.